# Revitalize Region Feature for Democratizing Video-language Pre-training of Retrieval

## Abstract

Recent dominant methods for video-language pre-training (VLP) learn transferable representations from the raw pixels in an end-to-end manner to achieve advanced performance on downstream video-language retrieval. Despite the impressive results, VLP research becomes extremely expensive with the need for massive data and a long training time, preventing further explorations. In this work, we revitalize region features of sparsely sampled video clips to significantly reduce both spatial and temporal visual redundancy towards democratizing VLP research at the same time achieving state-of-the-art results. Specifically, to fully explore the potential of region features, we introduce a novel bidirectional region-word alignment regularization that properly optimizes the fine-grained relations between regions and certain words in sentences, eliminating the domain/modality disconnections between pre-extracted region features and text. Extensive results of downstream video-language retrieval tasks on four datasets demonstrate the superiority of our method on both effectiveness and efficiency, *e.g.*, our method achieves competing results with 80% fewer data and 85% less pre-training time compared to the most efficient VLP method so far (Lei et al., 2021).

## 1 Introduction

Video-language pre-training (VLP) (Lei et al., 2021; Li et al., 2020a; Miech et al., 2020) that jointly learns video and language representations in a self-supervised manner has become the most popular practice to cope video-language retrieval (Lee et al., 2018; Liu et al., 2019a). Recently, end-to-end methods (Bain et al., 2021; Zellers et al., 2021) that learn video representations from raw pixels have dominated due to their strong performance on downstream tasks. Despite significant progress, these methods are quite data-hungry due to a large number of model parameters and uncurated raw inputs. The pre-training stage turns out to be inefficient and expensive with massive pre-training data and long pre-training time, making it difficult for researchers to pursue research in VLP.

Previous work (Lei et al., 2021) attempts to lower the barrier for VLP via removing visual redundancy. They point out that video clips with sparsely sampled frames are sufficient enough to capture key semantics for pre-training, since adjacent frames often contain similar scenes. The effort enables more efficient VLP with competitive downstream performances. Besides the temporal visual redundancy, we argue that, in contrast to the text with highly abstract semantics, each frame of the video clips also has heavy spatial redundancy.

Towards this end, we further propose to remove the redundant spatial information in sparsely sampled video clips via the claim that *a frame is actually worth around 30 objects* (based experiments in Section 4.4). Specifically, we revitalize offline region features that were all the rage in image-language tasks (Liu et al., 2019a) to encourage efficient VLP. Region features are generally pre-extracted by a pre-learned object detector (Anderson et al., 2018). Rather than the dense and continuous visual signal of the raw pixels, the region features are sparsely distributed with the compact information of salient visual contents, which are the most useful for video-text understanding. The sparse sampling significantly reduce the complexity of attention mechanism, which enables our model to have larger capacity with less FLOPs. We further advocate "*less is more*" for one more step towards democratizing VLP research.

As is known, methods using off-the-shelf features (Lee et al., 2018) have been phased out in visual-language tasks due to the inferior downstream performances. Previous work (Lei et al., 2021) at-

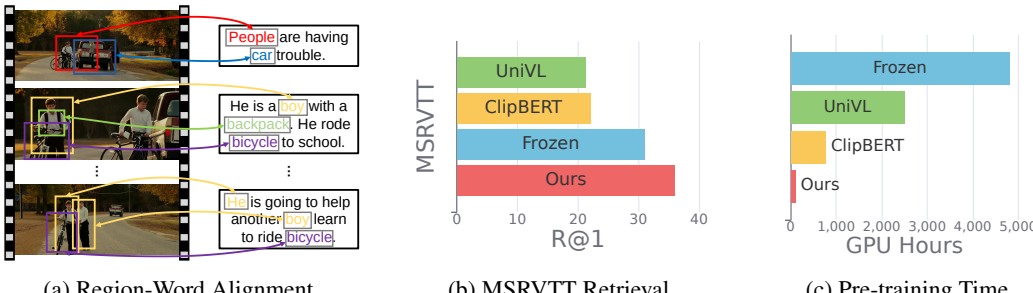

(a) Region-Word Alignment.     (b) MSRVTT Retrieval.     (c) Pre-training Time.

Figure 1: Region-word alignment and results. (a) Region-word alignment is to reason the detailed correspondence between salient regions and words. (b) and (c) demonstrate that our method significantly improves text-to-video retrieval while reducing pre-training time to a large extent.

tributes the unsatisfactory pre-training performance of pre-extracted features to their disconnections with the current domain and language modality. We would like to clarify that such disconnections can be properly eliminated by imposing fine-grained cross-modality alignment regularization.

Specifically, besides the common late fusion regularization on the global visual-text representations (Bain et al., 2021), we introduce a novel bidirectional region-word alignment regularization under the observation that objects extracted from video frames are naturally associated with certain words in the corresponding sentences. For instance, as demonstrated in Fig. 1, the keywords "people", "car" and "bicycle" share high-level semantics with cropped regions (highlighted with bounding boxes), respectively. To model and promote such a detailed cross-modality relationship, we build bidirectional connections between extracted regions and words. In the Region→Word manner, we estimate the region-to-sentence similarity resorting to the similarities between each region and all the words in a sentence. The average region-to-sentence similarity over all the regions of a video clip is treated as the video-to-sentence similarity, which is further maximized for positive pairs. Similarly, the Word→Region manner is conducted to measure and optimize the sentence-to-video similarity according to the similarities between each word and the corresponding regions. We surprisingly find that the proposed fine-grained region-word alignment constraints can also be seamlessly integrated into end-to-end VLP methods (Bain et al., 2021), achieving promising performance gains.

In summary, our contributions are three-fold: (1) We revitalize region features towards democratizing VLP via removing both temporal and spatial visual redundancy. Specifically, our efficient VLP model can maintain state-of-the-art performance on multiple downstream tasks with 80% fewer data and 85% less pre-training time than ClipBERT, which is the most efficient end-to-end VLP method so far. (2) We clarify that the inferior performance of off-the-shelf features in previous attempts (Li et al., 2020a; Zhu & Yang, 2020; Sun et al., 2019; Yu et al., 2018; Gabeur et al., 2020) lies in the sub-optimal learning regularization. We tackle the challenge with a newly proposed bidirectional region-word constraint, which optimizes fine-grained visual-text relations and properly eliminates the domain/modality disconnections of the region features. (3) Our method shows competitive results on four downstream video-language retrieval tasks. We surprisingly observe that the introduced region-word alignment regularization can also effectively boost the end-to-end method (Bain et al., 2021) with noticeable improvements.

## 2 RELATED WORK

**Video-Language Pre-training.** Early VLP methods (Li et al., 2020a; Zhu & Yang, 2020; Sun et al., 2019; Yu et al., 2018; Gabeur et al., 2020) introduce pretrained models on other tasks to pre-extract video representations. Some of them (Li et al., 2020a; Zhu & Yang, 2020; Sun et al., 2019) utilize action recognition backbones (Feichtenhofer et al., 2019; Hara et al., 2018) to pre-extract video representations. These backbones are designed with 2D (He et al., 2016) and 3D (Hara et al., 2018) CNNs to capture spatial and temporal information in videos. Others (Yu et al., 2018; Liu et al., 2019b; Gabeur et al., 2020; Wang et al., 2021b) fuse multiple "Experts" that are trained on different modalities, such as audio classification (Hershey et al., 2017), OCR (Gabeur et al., 2020), image classification (Huang et al., 2017) and so on, to fully exploit cross-modal high-level semantics in videos. Recently, end-to-end models (Miech et al., 2020; Lei et al., 2021; Bain et al., 2021; Zellers et al., 2021; Fu et al., 2021) are proposed . Some (Miech et al., 2020; Lei et al., 2021; Zellers et al., 2021) utilize CNNs to extract video features, others (Bain et al., 2021; Fu et al., 2021) replace CNNs with ViT (Dosovitskiy et al., 2021) to build a pure Transformer-based VLP model.

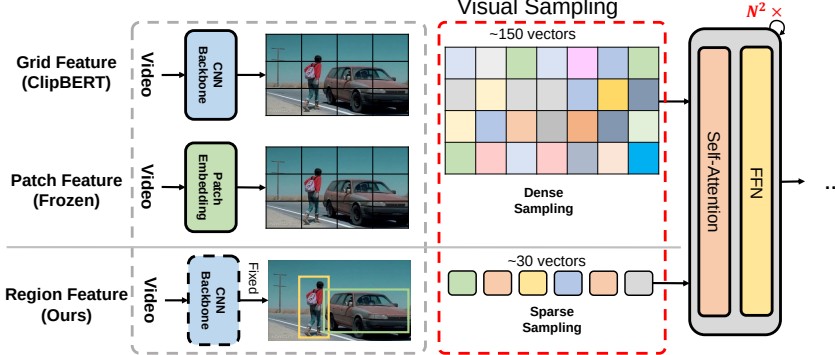

Figure 2: Comparison of conventional VLP visual sampling and our method. We introduce region feature to the VLP pipeline to achieve sparse sampling and accelerate the pre-training period.

**Region Features in Image-Language Pre-training.** Region features extracted by an object detector (Anderson et al., 2018) are adopted by image-language pre-training (ILP) methods (Chen et al., 2020b; Li et al., 2020b). They represent an image to a set of region features that focus on salient regions that are a much more natural basis for attention (Egly et al., 1994; Scholl, 2001). Region features enable the interaction between image and text at the object level. Recent ITP methods (Li et al., 2020b; Chen et al., 2020b; Yao et al., 2022) attempt to form fine-grained alignment between image and text based on region features.

**Fine-Grained Alignment.** Fine-grained alignment between visual and textual contents has been explored in image-language retrieval for several years. Some studies (Lee et al., 2018; Liu et al., 2019a; Li et al., 2017; Chen et al., 2020a; Wang et al., 2019) propose different attention mechanisms to match visual regions and captions. Recent ITP methods (Li et al., 2020b; Chen et al., 2020b) also explore the alignment between regions and words. Considering the low efficiency for video processing, previous VLP methods (Luo et al., 2020; Sun et al., 2019; Gabeur et al., 2020) adopt simple contrastive loss or binary classification to align video and language. Recently, TACo (Yang et al., 2021) has attempted to build up a fine-grained alignment. However, it still suffers from so large computation cost that simplifies its original design. Benefiting from our efficient pipeline, we propose a novel region-word alignment method to better connect the video and language.

## 3  METHOD

A video and its corresponding caption are taken as input. The video is initially embedded to region features by an off-the-shelf detector (Anderson et al., 2018) before being fed into a video encoder. The corresponding caption is encoded to language features by a language encoder. During the pre-training, the video and caption features are optimized towards the objective of global-local alignment. After pre-training, the model can be transferred to video-language retrieval with fine-tuning.

### 3.1  MODEL ARCHITECTURE.

**Input.** Given paired video $V$ and sentence $T$ as raw inputs. Similar to previous VLP methods (Bain et al., 2021; Lei et al., 2021), $T$ is tokenized into word tokens $\{w_l\}_{l=0}^{L}$, where $L$ denotes the number of tokens in $T$ and $w_0$ denotes the [CLS] token. These word tokens are inputted to language encoder. Video frames $\{f_1, f_2, ..., f_M\}$ are sampled from $V$, where $M$ denotes the number of frames. Given frames $\{f_1, f_2, ..., f_M\}$ of a video, objects and salient regions in each frame are detected by Faster RCNN (Anderson et al., 2018) pre-trained on Visual Genome (Krishna et al., 2017). Pooled RoI features of these regions are concatenated together as features $\{o_n \in \mathbb{R}^d\}_{n=1}^{N}$ of the whole video, where $N$ denotes the number of regions. Video encoder takes these RoI features as the inputs.

In this work, we use pre-extracted region features rather than fine-tuning the detector end-to-end for efficient pre-training. Tuning the detector requires extra computational and memory overhead, and it is also infeasible to acquire bounding box annotations in video-language datasets.

**Temporal Modeling.** We add a [CLS] token $o_0$ to represent the whole video so that the input becomes $\{o_n\}_{n=0}^{N}$. We combine temporal position embeddings and location embeddings to model temporal clues in videos. To encode location information for each region feature, we use a linear layer $FC$ to project 7-dimensional location vectors $\{l_n = [x_1, y_1, x_2, y_2, w, h, w * h]\}_{n=0}^{N}$ (normal-

ized top/left/bottom/right coordinates, width, height and area) to $\{l'_n \in \mathbb{R}^d\}_{n=0}^N$:

$$l'_n = FC(l_n) \tag{1}$$

The change of location reveals movement of an object. To indicate frames at different moment, we introduce learned temporal position embeddings $\mathbf{P} \in \mathbb{R}^{M \times d}$. Thus, region features with temporal clues are revised as $o'_n = o_n + l'_n + \mathbf{P}_m$, where $m$ denotes $o_n$ is extracted from $m$-th frame.

Compared with video transformer (e.g., TimeSformer (Bertasius et al., 2021)), our method captures temporal clues more efficiently by removing time attention and reaches comparable performance. Comparison of complexity and performance is described in Section 3.4 and Appendix A.4.5.

**Video Encoder.** Video Encoder encodes $\{o'_n\}_{n=0}^N$ with a Transformer-based network $E_V$:

$$\{r_n\}_{n=0}^N = E_V(\{o'_n\}_{n=0}^N) \tag{2}$$

where $r_n$ denotes the output region feature. We introduce ViT (Dosovitskiy et al., 2021) as $E_V$. Video Encoder makes region features interplay with each other, and outputs refined region features $\{r_n\}_{n=0}^N$, where $r_0$ is encoded from [CLS] token and treated as the global video feature.

**Language Encoder.** Similar to previous VLP methods (Lei et al., 2021; Bain et al., 2021), Language Encoder is based on BERT (Devlin et al., 2018). Word tokens are encoded to $\{t_l \in \mathbb{R}^d\}_{l=0}^L$:

$$\{t_l\}_{l=0}^L = E_V(\{w_l\}_{l=0}^L) \tag{3}$$

where $t_l$ denotes the output language feature and we adopt DistillBERT (Sanh et al., 2019) as $E_L$. $\{t_l\}_{l=0}^L$ are then aligned with region features by our proposed pre-training objective.

## 3.2 REDUCE VISUAL REDUNDANCY.

To democratize video-language pre-training, we attempt to remove both temporal and spatial redundancy. We involve two training strategies to ensure the pre-training time is acceptable.

**Temporal Visual Redundancy - Frame Sparse Sampling.** To save pre-training cost and ensure the performance on downstream tasks, we adopt a frame sapling strategy similar to ClipBERT (Lei et al., 2021). During the pre-training period, only a single frame is sampled randomly for each video. When finetuning on downstream tasks, a dense sampling strategy is adopted to capture more visual information from videos. We observe that such extremely sparse sampling still achieves competing results on downstream tasks in our framework. Experiments in Section 4.4 show how frame sampling strategy affects pre-training and finetuning.

**Spatial Visual Redundancy - Object Region Extraction.** Considering the number of objects and salient regions detected by a detector is uncertain, to balance the efficiency and performance, we explore strategies to determine the number of objects per frame and how to choose objects. To determine the number of objects, we conduct an ablation study and find that 30 objects per frame is the most suitable setting. Because it achieve performance close to more objects and less objects cause a obvious performance drop (Section 4.4).

As shown in Fig.2, compared with methods using dense sampling strategy (e.g., Frozen (Bain et al., 2021) with patch embeddings, ClipBERT (Lei et al., 2021) with grid features), our method utilize sparse feature vectors to represent a frame. Due to self-attention in Transformer is $O(N)^2$ time complexity, our method achieve much better efficiency.

## 3.3 PRE-TRAINING OBJECTIVE.

To jointly learn video-language representations from large-scale datasets (Bain et al., 2021), we introduce a global-local video-language alignment to connect video and language from coarse to fine. The global-local alignment consists of two pre-training objectives: **i) video-sentence alignment**, and **ii) region-word alignment**. For video-sentence alignment, we employ contrastive learning with [CLS] tokens to align video and language globally. In the region-word alignment, regions and words are aligned to reach fine-grained alignment via a newly proposed bidirectional regularization.

**Video-Sentence Alignment.** Two losses are optimized to increase the video-to-language and language-to-video similarity:

$$\mathcal{L}_{\text{v2l}}^{\text{global}} = -\frac{1}{B}\sum_i^B \log \frac{\exp(r_{\text{cls}}^{i^T} t_{\text{cls}}^i / \sigma)}{\sum_j^B \exp(r_{\text{cls}}^{i^T} t_{\text{cls}}^j / \sigma)} \tag{4}$$

$$\mathcal{L}_{\text{l2v}}^{\text{global}} = -\frac{1}{B} \sum_{i}^{B} \log \frac{\exp(t_{\text{cls}}^{i^T} r_{\text{cls}}^i / \sigma)}{\sum_{j}^{B} \exp(t_{\text{cls}}^{i^T} r_{\text{cls}}^j / \sigma)} \tag{5}$$

where $r_{\text{cls}}^i$ and $t_{\text{cls}}^j$ are normalized embeddings of $i$-th video's [CLS] token and $j$-th caption's [CLS] token in a batch of size $B$, respectively. $\sigma$ denotes the temperature coefficient.

**Region-Word Alignment.** The local alignment also attempts to maximize video-to-language and language-to-video similarity, but in a fine-grained way. We take video-to-language as an example to illustrate the details of region-word alignment. For $i$-th video with region features $\{r_n\}_{n=1}^N$ and $j$-th sentence with word features $\{t_l\}_{l=1}^L$, $n$-th region feature firstly attends to each word feature to pick up the most relevant words according to attention weights:

$$a_{n,l} = \frac{\exp(\langle r_n, t_l \rangle)}{\sum_{k=1}^{L} \exp(\langle r_n, t_k \rangle)} \tag{6}$$

where $\langle \cdot, \cdot \rangle$ denotes the cosine similarity, $a_{n,l}$ denotes the similarity between the $n$-th region feature and $l$-th word feature. Intuitively, semantics in videos are usually associated with nouns, verbs and adjectives Yang et al. (2021); Liu et al. (2019a). Other types of words, e.g., prepositions, are not "concrete" enough in videos Yang et al. (2021). Thus, to make the region only attend to the most relevant words and ignore other noisy words, we refine $a_{n,l}$ as follows:

$$a'_{n,l} = \mathbb{1}[a_{n,l} - \frac{1}{L} \sum_{k=1}^{L} a_{n,k}] \cdot a_{n,l} \tag{7}$$

where $\mathbb{1}[x] = 1$ if $x > 0$ else 0. $a'_{n,l}$ is the refined similarity between the $n$-th region feature and $l$-th word feature. By regarding the average of attention weights as threshold, we set attention weights of irrelevant words (below threshold) to 0. By conducting this operation, we ensure the model only focuses on the relevant words with respect to a video.

Given the refined attention weights, an attended sentence feature with respect to $n$-th region feature is calculated as: $\alpha_n = \sum_{l=1}^{L} a'_{n,l} t_l$, where $\alpha_n$ denotes the attended sentence feature. We compute such attended sentence feature for each region feature. By summing up the similarity between all region features and their corresponding attended sentence features, we get the similarity of $i$-th video and $j$-th caption:

$$S_{i,j} = \frac{1}{N} \sum_{n=1}^{N} \langle r_n, \alpha_n \rangle \tag{8}$$

The video-to-language loss is calculated with contrastive loss:

$$\mathcal{L}_{\text{v2l}}^{\text{local}} = -\frac{1}{B} \sum_{i}^{B} \log \frac{\exp(S_{i,i} / \sigma)}{\sum_{j}^{B} \exp(S_{i,j} / \sigma)} \tag{9}$$

where $\sigma$ denotes the temperature coefficient. The language-to-video loss $\mathcal{L}_{\text{l2v}}^{\text{local}}$ is similar to $\mathcal{L}_{\text{v2l}}^{\text{local}}$. More details are described in Appendix A.1.

**Overall Pre-training Objective.** Combining the above losses, the overall pre-training objective is: $\mathcal{L} = \mathcal{L}_{\text{v2l}}^{\text{global}} + \mathcal{L}_{\text{l2v}}^{\text{global}} + \mathcal{L}_{\text{v2l}}^{\text{local}} + \mathcal{L}_{\text{l2v}}^{\text{local}}$. The proposed alignment can fully exploit previous knowledge learned by region features and build up a more precised alignment compared with existing alignments (Yang et al., 2021; Yao et al., 2022) (We examine the effectiveness and analyze the reasons in Section 4.2).

## 3.4 COMPLEXITY ANALYSIS.

Previous offline-feature-based VLP methods (Li et al., 2020a; Zhu & Yang, 2020) are criticized for excessive demand on memory and computation. However, we claim that our method is more efficient than recent end-to-end methods (Bain et al., 2021; Lei et al., 2021) overall. Although we need to pay extra time to extract region features, it is negligible compared with the pre-training time that our method saves, especially we only need to extract them once but can reuse them for plenty of times. When it comes to downstream tasks, the detector only needs to extract features for queries, as we can assume gallery features are pre-extracted and stored in practical. Such extra time is also acceptable considering the performance improvements. We give a complexity analysis as follow.

**Pre-training.** We compare the pre-training time and R@1 on MSRVTT retrieval of our method with different settings and other methods. Because different methods use different number of GPUs, we

| Method | Data | GPU Hrs | R@1 |
|--------|------|---------|-----|
| Frozen | 5.8M | 4800 | 31.0 |
| UniVL | 132M | 2496 | 21.2 |
| HERO | 7.6M | 8064 | 20.5 |
| VIOLET | 185.8M | 2240 | 34.5 |
| ClipBERT | 5.6M | 768 | 22.0 |
| Ours (4F/1.0) | 5.8M | 1600 | 36.3 |
| Ours (1F/1.0) | 5.8M | 800 | 36.0 |
| Ours (1F/0.5) | 2.9M | 416 | 36.4 |
| Ours (1F/0.2) | 1.2M | 104 | 34.6 |

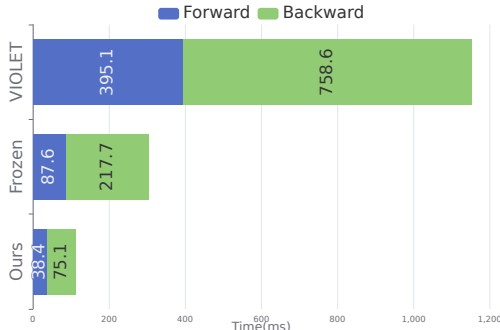

Table 1: Comparing the pre-training efficiency with existing video-language pre-training methods. 4F means that 4 frames per video are sampled for pre-training. 0.2 means that only 20% pre-training data are used.

Figure 3: Comparing the running time of a training loop with existing video-language pre-training methods. Batchsize is set to 16 and mixed precision is disabled for all methods.

introduce GPU Hours, i.e., *number of GPUs × pre-training time*, to measure the computational cost. In Table 1, our method is significantly more efficient than other methods. Our method achieves the best performance, even when we only use 20% pre-training data and extremely sample only 1 frame per video. Compared with the most efficient VLP method so far, i.e., ClipBERT (Lei et al., 2021), **our method save 85% pre-training time**. The reason why the model pretrained with 50% data performs better than those pretrained with more data is that finetuning results may not be strictly correlated with the data scale. Zero-shot results reflect the impact of data scale more precisely. In Fig. 4 and 6, zero-shot results degrade slightly as the size of data and the number of frames decreases.

Besides the whole pre-training, we also explore the running time of a training loop. We compare our method with Frozen (Bain et al., 2021) and VIOLET (Fu et al., 2021) to show the efficiency of our method. As shown in Fig. 3, our method requires much less time than Frozen (Bain et al., 2021) and VIOLET (Fu et al., 2021). The forward time of our method is only 43.8% of Frozen's and 9.7% of VIOLET's. The backward time of our method is only 34.5% of Frozen's and 9.9% of VIOLET's.

**Downstream Retrieval.** We test the inference time of each model components on text-to-video retrieval. We build up two galleries: **i)** test split of LSMDC that contains 1000 samples; **ii)** 12000 samples collected from WebVid's train split. In practical, samples in gallery are pre-extracted, thus we record the running time of detection, feature extraction of a query and ranking.

As shown in Table 2 (T→V denotes Text-to-Video), detection costs around 100ms for a video. Feature extraction of texts and videos takes very little time. As the gallery expands, ranking time becomes the major consumption. Considering the gallery capacity is usually larger than 12000 in practical, the extra time for detection is acceptable.

| Parts | 12000 Samples | | 1000 Samples | |
|-------|---------------|---------------|---------------|---------------|
| | T→V | V→T | T→V | V→T |
| Language | 5.6ms | - | 4.8ms | - |
| Video | - | 17.6ms | - | 16.4ms |
| Detector | - | 103.2ms | - | 103.2ms |
| Rank | 372.2ms | 345.8ms | 31.8ms | 30.0ms |

Table 2: Inference time of each component.

| Parts | Frozen | | Ours | |
|-------|--------|-------|------|-------|
| | FLOPs | Param | FLOPs | Param |
| Language | 9.7B | 66M | 9.7B | 66M |
| Video | 590B | 121.4M | 41.8B | 86.8M |
| Detector | - | - | 455.2B | 60.6M |
| All | 599.7B | 187.4M | 506.7B | 213.4M |

Table 3: Inference FLOPs of each component.

In Table 3, We compare our method with Frozen on inference FLOPs and parameters of each component. Frozen introduces TimeSformer as the video encoder. The dense visual sampling and time attention in TimeSformer increase the inference FLOPs rapidly. Thus, **our method even has lower inference FLOPs with more model parameters**.

## 4 EXPERIMENTS

### 4.1 DATASETS

**Pre-training.** Following the recent works (Bain et al., 2021; Fu et al., 2021), we involve a video-language dataset and an image-language dataset to pre-train our model. **i) WebVid2.5M (WebVid)**; and **ii) Google Conceptual Captions (CC3M)**.

| Method | Text→Video | | |
|---|---|---|---|
| | R@1 | R@5 | R@10 |
| Base | 22.5 | 48.1 | 59.5 |
| Ours(Base+RWA) | 36.0 | 61.0 | 71.8 |
| Zero-shot | | | |
| Base | 16.1 | 34.9 | 44.2 |
| Ours(Base+RWA) | 22.5 | 48.1 | 59.5 |

(a) Results on MSRVTT retrieval with Base.

| Method | Text→Video | | |
|---|---|---|---|
| | R@1 | R@5 | R@10 |
| Frozen | 31.0 | 59.5 | 70.5 |
| Frozen+RWA | 35.1 | 60.7 | 70.9 |
| Zero-shot | | | |
| Frozen | 18.7 | 39.5 | 51.6 |
| Frozen+RWA | 21.7 | 42.2 | 52.2 |

(b) Results on MSRVTT retrieval with Frozen.

Table 4: Effectiveness of our region-word alignment. "Base" denotes the baseline that adopts the same region features as input while only using video-sentence global alignment as the objective.

| Method | Text→Video | | |
|---|---|---|---|
| | R@1 | R@5 | R@10 |
| Base+TACo | 29.8 | 60.1 | 70.7 |
| Base+FILIP | 27.8 | 55.0 | 65.1 |
| Ours(Base+RWA) | **36.0** | **61.0** | 71.8 |

| Method | Text→Video | | |
|---|---|---|---|
| | R@1 | R@5 | R@10 |
| Ours w/o Refinement | 34.5 | 58.3 | 70.2 |
| Select Token | 32.8 | 55.5 | 69.2 |
| Ours | 36.0 | 61.0 | 71.8 |

Table 5: Comparison to other alignments. "Base" is the model whose pre-train objective only contains global alignment.

Table 6: Ablation study of refinement. Select Token denotes that input captions only contain nouns and verbs.

**Downstream Tasks.** We evaluate our method on text-to-video retrieval, across 4 downstream tasks. We report results on *i)* **MSRVTT**; *ii)* **DiDeMo**; *iii)* **LSMDC** and *iv)* **MSVD**. We adopt the common R@K (K=1, 5, 10) metric and Median Rank (MedR) to measure the performance of retrieval.

Details of datasets and implementation are described in Appendix A.2 and A.3.

## 4.2 ANALYSIS OF REGION-WORD ALIGNMENT

In this section, we verify the effectiveness of region-word alignment (RWA) by compare with a baseline model and other alignment methods. Furthermore, by analyzing experimental results, we examine that region-based models require fine-grained alignment to exploit prior learned by region features, and an appropriate mechanism is essential for precise alignment.

**Effectiveness.** We conduct an ablation study of RWA on Base (i.e., the pre-training objective is modified into $\mathcal{L} = \mathcal{L}_{v2l}^{global} + \mathcal{L}_{l2v}^{global}$) and Frozen. Results on MSRVTT retrieval are shown in Table 4.

It's obvious that RWA improve the performance of Base and Frozen on all metrics for finetuning and zero-shot settings. The results demonstrate that RWA can better align language and videos and thus enhance the performance of VLP. Another advantage of RWA is that it can adapt to end-to-end methods (e.g., Frozen) seamlessly as shown in Table 4b.

**Comparison with other alignment methods.** We compare RWA with other fine-grained alignment methods, e.g., TACo (Yang et al., 2021) and FILIP (Yao et al., 2022), on MSRVTT retrieval. Resutls are shown in Table 5. Though they somewhat eliminate the disconnections between region features and language modality with improved performance over baseline, they are still much inferior to ours since they only associate the image/region with a single word. However, a region can correspond to multiple words, and these cases are covered by RWA. The above sub-optimal alignment is not critical for their end-to-end frameworks due to the incremental gains, but is fatal to our framework due to the incomplete elimination of disconnections, leaving a significant gap from SOTA results.

**Ablation on Refinement.** We refine attention weights in Eq (7) to choose the most relevant words and avoid noisy words hurting the alignment between words and regions. To determine the effect of this operation, we compare it with another operation, i.e., **Select Token**, which manually selects nouns and verbs as the input caption. Because nouns and verbs usually contain more semantics than other types of words (Yang et al., 2021). As shown in Table 6, our proposed refinement improve results of the model without refinement. **Select Token** even hurts the performance of baseline. It means that filtering out noisy words can indeed help to align words with regions, however, an appropriate implement is important.

**Discussion.** As is known, region features extracted by a detector have already learned the concepts of different "objects". Utilizing such prior to better align video and language is the intuition of our region-word alignment. In Table 4, our region-based model performs worse than end-to-end

| Method | Text→Video | | | |
|---|---|---|---|---|
| | R@1 | R@5 | R@10 | MedR |
| JSFusion | 9.1 | 21.2 | 34.1 | 36.0 |
| MEE | 9.3 | 25.1 | 33.4 | 27.0 |
| CE | 11.2 | 26.9 | 34.8 | 25.3 |
| MMT | 12.9 | 29.2 | 38.8 | 19.3 |
| AVLNet | 17.0 | 38.0 | 48.6 | 11.0 |
| Dig | 15.8 | 34.1 | 43.6 | 14.3 |
| Frozen | 15.0 | 34.1 | 39.8 | 20.0 |
| VTMCE | 14.9 | 33.2 | - | 19.0 |
| MDMMT | 18.8 | 38.5 | 47.9 | 12.3 |
| **Ours** | **25.2** | **45.5** | **54.5** | **7.0** |
| Zero-shot | | | | |
| Ours | 14.3 | 25.8 | 32 | 49.5 |

(a) LSMDC retrieval

| Method | Text→Video | | | |
|---|---|---|---|---|
| | R@1 | R@5 | R@10 | MedR |
| MMT | 26.6 | 57.1 | 69.6 | 4.0 |
| ActBERT | 16.3 | 42.8 | 56.9 | 10.0 |
| SupportSet | 30.1 | 58.5 | 69.3 | 3.0 |
| AVLNet | 27.1 | 55.6 | 66.6 | 4.0 |
| TACo | 29.6 | 59.7 | **72.7** | 4.0 |
| ClipBERT | 22.0 | 46.8 | 59.9 | 6.0 |
| Frozen | 31.0 | 59.5 | 70.5 | 3.0 |
| **Ours** | **36.0** | **61.0** | 71.8 | **3.0** |
| Zero-shot | | | | |
| SupportSet | 12.7 | 27.5 | 36.2 | 24.0 |
| Frozen | 18.7 | 39.5 | 51.6 | 10.0 |
| **Ours** | **24.0** | **44.0** | **52.6** | **8.0** |

(b) MSRVTT retrieval

| Method | Text→Video | | | |
|---|---|---|---|---|
| | R@1 | R@5 | R@10 | MedR |
| S2VT | 11.9 | 33.6 | - | 13.0 |
| FSE | 13.9 | 36.0 | - | 11.0 |
| CE | 22.6 | 51.1 | - | 5.0 |
| ClipBERT | 20.4 | 44.5 | 56.7 | 7.0 |
| Frozen | 31.0 | 59.8 | 72.4 | 3.0 |
| **Ours** | **41.4** | **67.6** | **77.6** | **2.0** |
| Zero-shot | | | | |
| Frozen | 21.1 | 46.0 | 56.2 | 7.0 |
| **Ours** | **29.6** | **53.0** | **65.1** | **4.0** |

(c) DiDeMo retrieval

| Method | Text→Video | | | |
|---|---|---|---|---|
| | R@1 | R@5 | R@10 | MedR |
| VSE | 12.3 | 30.1 | 42.3 | 14.0 |
| VSE++ | 15.4 | 39.6 | 53.0 | 9.0 |
| MCues | 20.3 | 47.8 | 61.1 | 6.0 |
| CE | 19.8 | 49.0 | 63.8 | 6.0 |
| SupportSet | 28.4 | 60.0 | 72.9 | 4.0 |
| Frozen | 33.7 | 64.7 | 76.3 | 3.0 |
| **Ours** | **50.9** | **78.9** | **87.0** | **1.0** |
| Zero-shot | | | | |
| Ours | 41.6 | 69.8 | 80.5 | 2.0 |

(d) MSVD retrieval

Table 7: Comparisons with state-of-the-art results on text-to-video retrieval.

model (i.e., Frozen) when only global alignment is conducted. However, region-based model outperforms end-to-end model when adding fine-grained alignment. The results verify that fine-grained alignment is exactly the appropriate way to exploit the implicit semantics in region features.

However, as shown in Table 5, existing fine-grained alignment improve the Base model with limited gains. It is attributed to the way these methods build up fine-grained alignment, where they only choose one word for a region. In real-world scenario, a region could be associated with multiple words. In our methods, we align a region with multiple words and vice versa. This modification enables our method build up a more precise alignment between video and language.

### 4.3 COMPARISONS WITH STATE-OF-THE-ART

We compare our method with state-of-the-art methods on text-to-video retrieval tasks. All results in this section is based on the model that samples 1 frame per video during pre-training and samples 8 frames per video during finetuning.

Table 7 summarizes the results on four benchmarks. Across all tasks, our method achieves significant improvement compared with previous methods in both finetuning and zero-shot settings. On LSMDC, our method outperforms MEE (Dzabraev et al., 2021) by 6.4 on R@1. Similarly, on MSRVTT, our method surpasses Frozen (Bain et al., 2021) by 5.0 and 5.3 on R@1 for finetuning and zero-shot settings. For DiDeMo and MSVD, our method brings more than 10 improvement compared with existing best methods. Another advantage of our method is that our zero-shot performance significantly surpasses other methods. This means that our method builds up a good alignment between video and language during pre-training and generalizes well on different datasets.

### 4.4 ABLATION STUDY

**Data Scale.** We explore the impact of different pre-training data scales. We compare models that are pre-trained with different scales of data, i.e., 20%, 50%, 60%, 80% and 100%, on MSRVTT retrieval. As shown in Fig. 4, as the data scale decreases to 50%, R@1 of different scales are very close. Only if the scale drops to 20%, R@1 decreases to 34.6, which is still better than other state-of-the-art methods' results, e.g., Frozen, SupportSet and TACo. The model without RWA has a similar trend to the one with RWA across different scales.

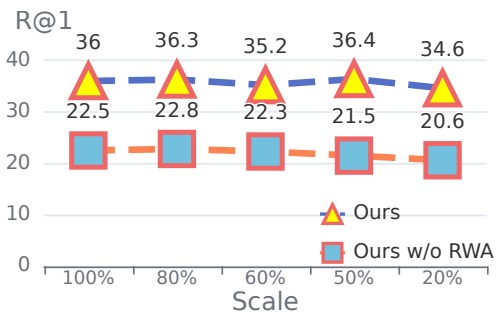

Figure 4: Effect of different data scales.

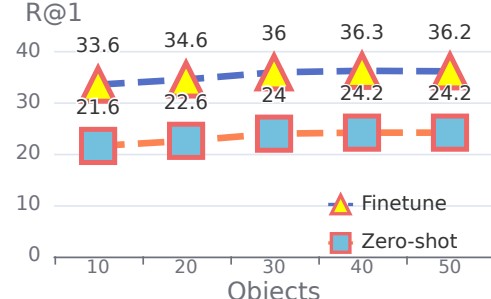

Figure 5: Different number of objects.

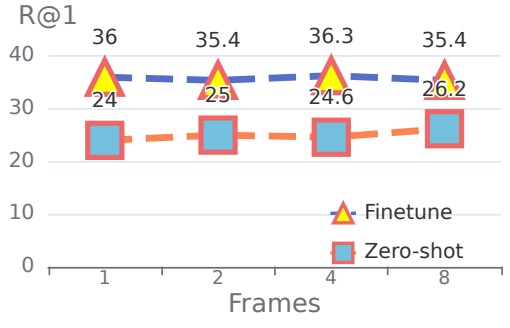

Figure 6: Number of frames (pre-training).

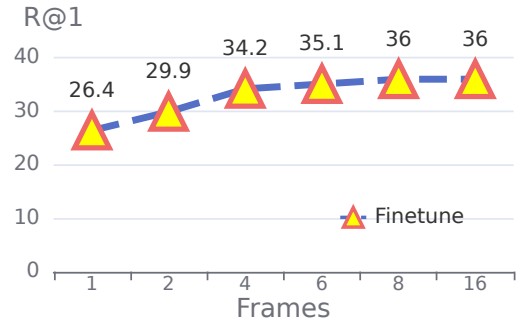

Figure 7: Number of frames (finetune).

**Number of Regions.** To determine how many regions can represent a video frame, we vary the number of regions and test our method's performance on MSRVTT retrieval. As the results shown in Fig. 5, R@1 is improved as the number of regions increases from 10 to 30. When the number continues to increase, the gain on R@1 is not obvious. Considering more regions increase time complexity rapidly, we generally represent a frame with 30 regions.

**Sampling Strategy.** To quantitatively evaluate the impact of sparsely sampling strategy during pre-training, we compare four models that sample 1, 2, 4, 8 frames per video respectively. When conducting downstream tasks, 8 frames are sampled. The results on MSRVTT retrieval are shown in Fig. 6. As the number of frames increases, the zero-shot results are improved, whereas the finetuning results are not closely related to the number of frames. The results show that dense sampling during pre-training is not necessary. This means that we can sparsely sampling frames during pre-training to save pre-training time.

Similarly, we evaluate the impact of sampling strategy during finetuning. We use the same pre-training model and then sample 1, 2, 4, 6, 8, 16 frames per video respectively during finetuning. The results on MSRVTT retrieval are shown in Fig. 7. Generally, as the sampling frames increase, the retrieval performance increases. When the number of frames is 16, compared with sampling 8 frames, no gain is achieved. Thus, 8 frames is enough for our model.

Results in Figs. 6 and 7 prove that sparse sampling during pre-training and dense sampling during finetuning is a reasonable solution for text-to-video retrieval.

## 5 CONCLUSION

In this work, we introduce the most efficient video-language pre-training method for video-language retrieval to date, which revitalizes region features with a bidirectional region-word alignment regularization. Region features that focus on salient areas remove spatial visual redundancy, which enables our VLP method to be extremely time-saving. The region-word alignment constraint builds up a fine-grained connection between video and language. Experimental results on downstream tasks and ablation studies prove the efficiency and effectiveness of our method.

**Limitation.** In our current implementation, we need a pre-trained detector to extract region features in advance. This step brings extra time consumption, which is not significant compared with saved time. However, we still acknowledge it is a limitation. A feasible extension is to use knowledge distillation to enable the model learn region features in an end-to-end manner. Despite the limitation, our method still provides an efficient paradigm to democratize VLP for researchers.

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

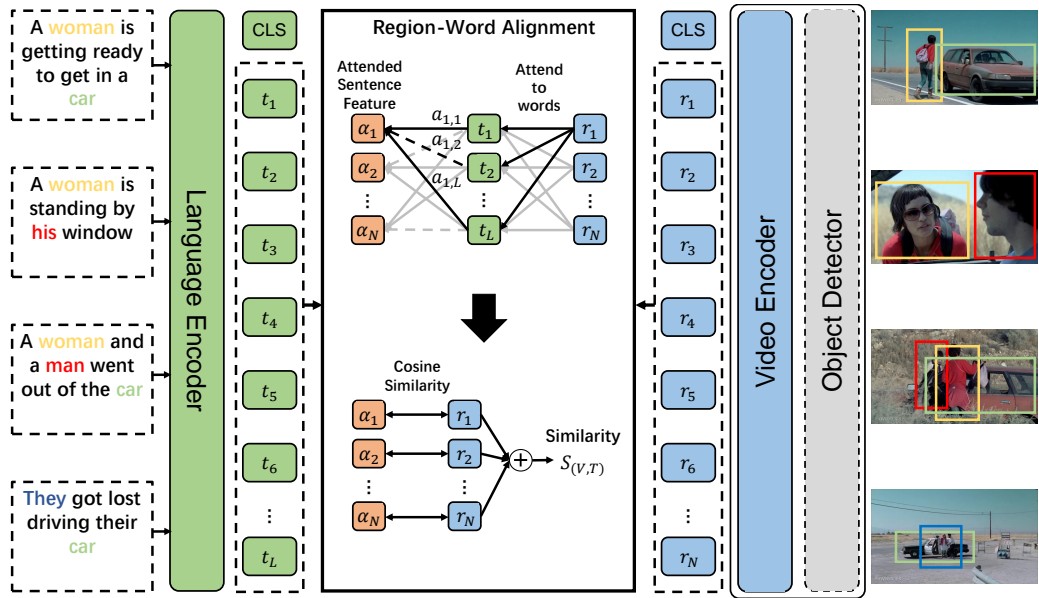

Figure 8: Overview of the region-word alignment. Given region features of a video and word tokens of a caption, we encode them via the video and language encoder respectively. Region-Word Alignment builds up a fine-grained connection between each region feature and word feature.

## A    APPENDIX

In this appendix, we first provide complete description of language-to-video loss. Then, we give more details of downstream tasks and implementation details. To show the generation of our framework, we give more additional experimental results, including video-to-text retrieval, video question answering and visualization.

### A.1    MORE DETAILS OF REGION-WORD ALIGNMENT.

In Fig. 8, we show the pipeline of region-word alignment. . A region feature firstly attends to word features to get attention weights of each word, and then fuse these word features to an attended sentence feature according to the weights. Each region corresponds to a specific attended sentence feature. Then, we compute the similarity between paired region features and sentence features. By summing up all these similarities, we get get the similarity between a video and sentence.

### A.1.1    LANGUAGE-TO-VIDEO LOSS.

We also illustrate the details of language-to-video loss. The language-to-video loss $\mathcal{L}_{\text{l2v}}^{\text{local}}$ is calculated in the similar way as $\mathcal{L}_{\text{v2l}}^{\text{local}}$. Given the similarity $a_{l,n}$ between $l$-th word and $n$-th region, the attended video vector with respect to $l$-th word is calculated:

$$\beta_l = \sum_{n=1}^{N} \mathbb{1}[a_{l,n} - \frac{1}{L}\sum_{n}^{N} a_{l,n}] \cdot a_{l,n} \cdot r_n \tag{10}$$

where $\beta_l$ denotes the attended video feature. The similarity of $j$-th caption and $i$-th video is:

$$S_{j,i} = \frac{1}{L}\sum_{l=1}^{L} \langle t_l, \beta_l \rangle \tag{11}$$

The language-to-video loss $\mathcal{L}_{\text{l2v}}^{\text{local}}$ is then given as follows:

$$\mathcal{L}_{\text{l2v}}^{\text{local}} = -\frac{1}{B}\sum_{j}^{B} \log \frac{\exp(S_{j,j}/\sigma)}{\sum_{i}^{B} \exp(S_{j,i}/\sigma)} \tag{12}$$

## A.2 DETAILS OF DATASETS.

**Pre-training. i) WebVid2.5M (WebVid)** consists of 2.5M video-language pairs collected from web. WebVid contains manually generated captions that are well-formed sentences and well-aligned with videos. **ii) Google Conceptual Captions (CC3M)** consists of 3.3M image-language pairs. Images and raw textual descriptions are harvested from the web following a similar process to WebVid.

**Text-to-Video Retrieval.** *i)* **MSRVTT** consists of 10K videos with 200K captions. We follow the previous works (Bain et al., 2021; Liu et al., 2019b) to train on 9K train and validation videos and test on the 1K test set. *ii)* **DiDeMo** consists of 10K videos with 40K captions. We concatenate all sentences of a video to a single query as previous works (Bain et al., 2021; Lei et al., 2021), and we do not use the ground-truth localization annotations of this dataset. *iii)* **LSMDC** contains 128K videos. We follow the setting in (Bain et al., 2021) to test our method on the 1K test set. *iv)* **MSVD** contains 1,970 videos. We split MSVD into 1,200, 100 and 670 videos as the train, validation and test set.

**Video Question Answering.** *i)* **MSRVTT Multiple Choice** is a question answering task that videos are questions and captions are answer candidates. Each video has 5 captions, and only caption is the positive one. *ii)* **MSRVTT-QA** is based on MSRVTT dataset. 243K open-ended questions and 1500 answers are annotated. *iii)* **MSVD-QA** is based on MSVD dataset. It contains a total number of 1,970 videos and 50,505 question-answer pairs. *iv)* **TGIF-FrameQA** is an open-ended QA tasks based on TGIF (Jang et al., 2017) dataset. TGIF (Jang et al., 2017) contains 165K QA pairs on 72K animated GIFs.

**Video Captioning.** *i)* **MSRVTT Caption** is video captioning task that requires the model to describe the visual content of a given video in natural language. The dataset consists of 10K open-domain video clips. Each video clip has 20 ground-truth captions. We use the standard captioning split, which has 6.5K training videos and 2.9K testing videos.

## A.3 IMPLEMENTATION DETAILS.

We implement our method with Pytorch and train all models on Tesla V100 GPUs with a batch size of 128 per GPU. We use the Adam optimizer with different learning rate schedule on pre-training and downstream finetuning. For pre-training, we train our model in 50 epochs using an initial learning rate of $1 \times 10^{-5}$. The learning rate decays to 1/10 of the previous one at 30 and 40 epochs. For fine-tuning, all experiments on downstream tasks are trained in 10 epochs. We densely sample 8 frames for each video. The initial learning rate is $1 \times 10^{-5}$ and decays to 1/10 at 2, 4 and 8 epochs. Detailed settings of each downstream task are as below.

**Text-to-Video Retrieval.** We sum up the video-sentence and region-word similarities as the final similarity between video and text. The final similarity is used to rank all video-language pairs.

The compared methods includes JSFusion (Yu et al., 2018), MEE (Miech et al., 2018), CE (Liu et al., 2019b), MMT (Gabeur et al., 2020), AVLNet (Rouditchenko et al., 2021), Dig (Wang et al., 2021b), Frozen (Bain et al., 2021), VTMCE (Ali et al., 2022), MDMMT (Dzabraev et al., 2021), ActBERT (Zhu & Yang, 2020), SupportSet (Patrick et al., 2021), TACo (Yang et al., 2021), Clip-BERT (Lei et al., 2021), S2VT (Venugopalan et al., 2014), FSE (Zhang et al., 2018), VSE (Kiros et al., 2014), VSE++ (Faghri et al., 2017), and MCues (Mithun et al., 2018).

**Multiple Choice.** For MSRVTT Multiple Choice, we directly use the model finetuned for MSRVTT retrieval to evaluate the performance. Specifically, we regard videos as questions and regard captions as answers. Each video contains 5 captions, and only one matches the video. By using the model pretrained for MSRVTT retrieval, we perform retrieval on each video and its corresponding 5 captions. The caption with the highest similarity is the predicted answer.

The compared methods includes JSFusion (Yu et al., 2018), ActBERT (Zhu & Yang, 2020), Clip-BERT (Lei et al., 2021), VideoCLIP (Xu et al., 2021), MERLOT (Zellers et al., 2021), VIOLET (Fu et al., 2021).

**Open-Ended Question Answering.** For MSRVTT-QA, MSVD-QA and TGIF-FrameQA that are open-ended question answering tasks, we adopt the framework in BUTD (Anderson et al., 2018), which uses a multi-label classifier based on region features and text features of questions to perform

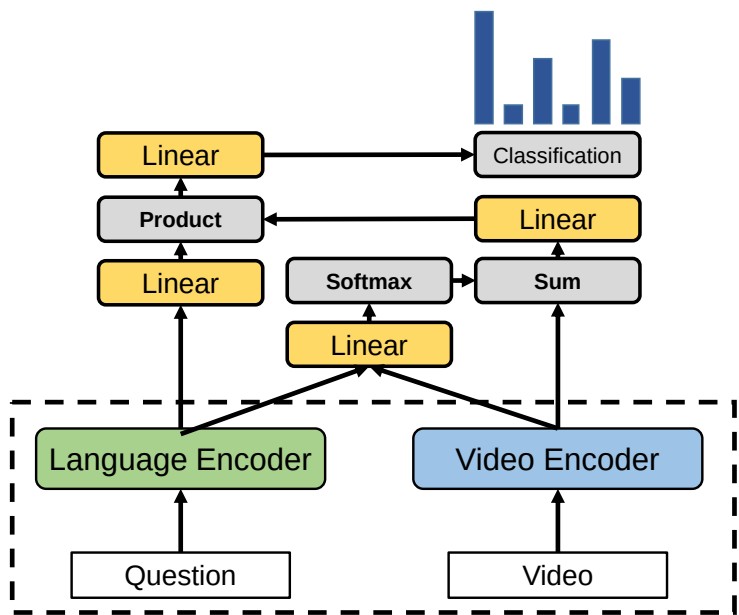

Figure 9: Overview of the proposed method applied to open-ended QA. We utilize a BUTD Anderson et al. (2018) head to fuse video and language features to perform open-ended QA tasks.

question answering task. As shown in Fig. 9, features of [CLS] from language encoder and region features from video encoder are fused by the BUTD head and conduct classification to choose a proper answer.

The compared methods include Co-Mem (Gao et al., 2018), HMEMA (Fan et al., 2019), SSML (Amrani et al., 2021), HCRN (Le et al., 2020), DualVGR (Wang et al., 2021a), Clip-BERT (Lei et al., 2021).

**Video Captioning.** Following the Language Encoder and Video Encoder, we build a Multi-modality Transformer Encoder (BERT-based) to further fuse word embeddings and object features in video. The training objective consists of two parts: optimizing (a) the global and local alignment loss, and (b) masked language modeling loss (by predicting masked tokens). We finetune our pretrained model on MSRVTT Caption dataset with 10epochs.

The compared methods include SupportSet (Patrick et al., 2021), UniVL (Luo et al., 2020), VideoAsMT (Korbar et al., 2020), OA-BTG (Zhang & Peng, 2019), GRU-EVE (Aafaq et al., 2019), SAAT (Zheng et al., 2020), and STG-KD (Pan et al., 2020). We provide results using a diverse set of performance metrics, including BLEU4, METEOR, ROUGE-L and CIDEr.

## A.4 SUPPLEMENTARY EXPERIMENTS.

To show the generalization of our method, we compare our method with the state-of-the-art results on video-to-text retrieval. Then, we test it on additional downstream tasks: **Video Question Answering**, including MSRVTT multiple choice, MSRVTT Question Answering, MSVD Question Answering and TGIF-FrameQA. Finally, we pretrain our method with more data to show how it generalize to a larger dataset and give some visualization results to show how region-word alignment is built.

### A.4.1 VIDEO-TO-TEXT RETRIEVAL

Considering vision-language is a bi-directional task, we compare our method with the same methods in Section 4.3 on video-to-text retrieval tasks, All results in this section is based on the model that samples 1 frame per video during pre-training and samples 8 frames per video during finetuning.

| Method | Video→Text | | | |
|---|---|---|---|---|
| | R@1 | R@5 | R@10 | MedR |
| JSFusion | - | - | - | - |
| MEE | - | - | - | - |
| CE | - | - | - | - |
| MMT | - | 28.6 | - | 20.0 |
| AVLNet | - | - | - | - |
| Dig | 14.3 | 33.7 | 43.6 | 15.5 |
| Frozen | - | - | - | - |
| VTMCE | 15.3 | 34.1 | - | 18.0 |
| MDMMT | - | - | - | - |
| **Ours** | **23.8** | **43.6** | **54.1** | **8.0** |
| Zero-shot | | | | |
| Ours | 16.0 | 29.3 | 35.5 | 40.5 |

(a) LSMDC retrieval

| Method | Video→Text | | | |
|---|---|---|---|---|
| | R@1 | R@5 | R@10 | MedR |
| MMT | 27.0 | 57.5 | 69.7 | 3.7 |
| ActBERT | - | - | - | - |
| SupportSet | 28.5 | **58.6** | 71.6 | **3.0** |
| AVLNet | - | - | - | - |
| TACo | - | - | - | - |
| ClipBERT | - | - | - | - |
| Frozen | - | - | - | - |
| **Ours** | **32.7** | 58.4 | 68.4 | 4.0 |
| Zero-shot | | | | |
| SupportSet | 8.7 | 23.0 | 31.1 | 31.0 |
| Frozen | - | - | - | - |
| **Ours** | **27.1** | **45.8** | **54.5** | **7.5** |

(b) MSRVTT retrieval

| Method | Video→Text | | | |
|---|---|---|---|---|
| | R@1 | R@5 | R@10 | MedR |
| S2VT | 13.2 | 33.6 | - | 15.0 |
| FSE | 13.1 | 33.9 | - | 12.0 |
| CE | 22.5 | 52.3 | - | 5.0 |
| ClipBERT | - | - | - | - |
| Frozen | - | - | - | - |
| **Ours** | **36.0** | **65.7** | **75.6** | **3.0** |
| Zero-shot | | | | |
| Frozen | - | - | - | - |
| Ours | 29.8 | 53.1 | 63.6 | 4.0 |

(c) DiDeMo retrieval

| Method | Video→Text | | | |
|---|---|---|---|---|
| | R@1 | R@5 | R@10 | MedR |
| VSE | 15.8 | 30.2 | 41.4 | 14.0 |
| VSE++ | 21.2 | 43.4 | 52.2 | 9.0 |
| MCues | 31.5 | 51.0 | 61.5 | 5.0 |
| CE | - | - | - | - |
| SupportSet | 28.7 | 60.8 | - | 2.0 |
| Frozen | - | - | - | - |
| **Ours** | **43.9** | **74.6** | **83.8** | **2.0** |
| Zero-shot | | | | |
| Ours | 41.3 | 67.8 | 78.3 | 2.0 |

(d) MSVD retrieval

Table 8: Comparisons with state-of-the-art results on video-to-text retrieval.

Table 8 summarizes the results on four benchmarks. Across all tasks, our method achieves significant improvement compared with previous methods in both finetuning and zero-shot settings. Considering our method also achieves an obvious advantage on text-to-video retrieval, the results prove that our method builds up a good alignment between video and language during pre-training and generalizes well on different datasets.

### A.4.2 VIDEO QUESTION ANSWERING.

| Method | MSRVTT |
|---|---|
| JSFusion | 83.4 |
| ActBERT | 85.7 |
| ClipBERT | 88.2 |
| VideoCLIP | 92.1 |
| MERLOT | 90.9 |
| VIOLET | 91.9 |
| **Ours** | **92.4** |

(a) MSRVTT Multiple Choice

| Method | MSRVTT | MSVD |
|---|---|---|
| Co-Mem | 32.0 | 31.7 |
| HMEMA | 33.0 | 33.7 |
| SSML | 35.0 | 35.1 |
| HCRN | 35.6 | 36.1 |
| DualVGR | 35.5 | 39.0 |
| ClipBERT | 37.4 | - |
| **Ours** | **38.3** | **39.5** |

(b) MSRVTT QA and MSVD QA

Table 9: Comparisons with state-of-the-art results on video question answering.

Table 9 summarizes the results on video question answering tasks. On MSRVTT MC, our method obtains accuracy of 92.4%, which outperforms prior state-of-the-art VideoCLIP (Xu et al., 2021) by 0.5%. On MSRVTT-QA and MSVD-QA, our method achieves accuracy of 38.3% and 39.5%. Compared to ClipBERT (Lei et al., 2021), our method brings 0.9% improvement on MSRVTT-QA. It also surpasses DualVGR (Wang et al., 2021a) by 0.5% on MSVD-QA.

| Method | TGIF-FrameQA |
|---|---|
| ST-VQA Jang et al. (2017) | 49.3 |
| Co-Mem Gao et al. (2018) | 51.5 |
| PSAC Li et al. (2019) | 55.7 |
| HMEMA Fan et al. (2019) | 53.8 |
| HCRN Le et al. (2020) | 55.9 |
| QueST Jiang et al. (2020) | 59.7 |
| ClipBERT Lei et al. (2021) | 60.3 |
| **Ours** | **60.6** |

Table 10: Comparisons with State-of-the-art results on TGIF-FrameQA.

### A.4.3 RESULTS ON TGIF-FRAMEQA.

FrameQA requires a model to highlight the fact that questions in this task can be answered according to a video. The answer comes from a dictionary of words of type object, number, color and location.

Results are shown in Table 10. Our method obtains accuracy of 60.6, which outperforms other methods. This experiment is a complementary for video QA tasks. The results also verify that our method works well on video QA tasks.

### A.4.4 REGION SELECTION.

To choose the most discriminative regions, we explore two selection strategies: **i) Sorted Selection**. Top 30 regions with the highest detection confidence are selected. **ii) Track Object**. We conduct object tracking to form a tracklet for each object. Only one region feature is reserved for each tracklet. We attempt to remove redundant objects that repeat in continuous frames to cover more object categories with this strategy.

To compare different region selection methods: **i) Sorted Selection** and **ii) Track Object**, we give their performance on MSRVTT retrieval. As shown in Table 11, **Sorted Selection** performs better than **Track Object** by 2.4 on R@1. Because **Sorted Selection** potentially choose multiple objects in the same category and **Track Object** attempts to cover more categories and only select one object per category, the results This means that discriminative region features verify that are more important for VLP than covering more object categories.

### A.4.5 TEMPORAL MODELLING.

Frozen utilizes TimeSformer (Bertasius et al., 2021) as the backbone, which proposes a time-attention layer to model temporal information. In our work, we rely on temporal position embeddings and location embeddings to model temporal information. We conduct ablation study on MSRVTT retrieval. As shown in Table 12, time-attention layer cannot bring improvement to our method. The results infer that, for VLP of retrieval, temporal position embeddings are enough to model temporal clues.

| Method | Text→Video | | |
|---|---|---|---|
| | R@1 | R@5 | R@10 |
| Track Object | 33.6 | 59.2 | 69.7 |
| Sorted Selection | 36.0 | 61.0 | 71.8 |

Table 11: Region selection.

| Method | Text→Video | | |
|---|---|---|---|
| | R@1 | R@5 | R@10 |
| Time attention | 35.9 | 60.8 | 71.8 |
| Temporal position | 36.0 | 61.0 | 71.8 |

Table 12: Temporal modeling.

### A.4.6 RESULTS WITH MORE DATA

Furthermore, to verify whether our method can generalize to larger datasets or not, we evaluate our method pre-trained with more data on MSRVTT retrieval. Specifically, we add a subset of Conceptual 12M (Changpinyo et al., 2021) as pre-training data. The subset contains 7M samples that are disjoint from CC3M, denoted as CC7M. As results shown in Table 13, CC7M can further improve the results of our method. It verifies that our method can extend to massive data to achieve better

| Data | Data Scale | Text→Video | | |
|---|---|---|---|---|
| | | R@1 | R@5 | R@10 |
| WebVid | 2.5M | 27.6 | 50.4 | 61.7 |
| WebVid+CC3M | 5.8M | 36.0 | 61.0 | 71.8 |
| CC3M+CC7M | 10.3M | 38.7 | 63.8 | **73.9** |
| WebVid+CC3M+CC7M | 12.8M | **39.6** | **64.8** | 73.8 |

Table 13: Results on MSRVTT retrieval with more data.

performance. An interesting observation is that adding data from original datasets (i.e., WebVid and CC3M) cannot improve the performance, while adding extra data from a different domain (i.e., CC7M) can. This observation inspires us that the diversity of data plays an important role in VLP.

### A.4.7 VISUALIZATION

We also provide qualitative results on Fig. 10 to show the alignment between video and language. In Fig. 10a, we visualize the attended regions with respect to the most relevant word in the sentence. We observe that not only nouns are connected with corresponding regions, even verbs, such as "see" and "smile", are also aligned with relevant regions. Fig. 10b shows the attention heatmap between video frames and corresponding sentence. Regions that are relevant to the sentence description show higher attention weights than others. Visualization verifies that our proposed global-local alignment successfully connects video and language from coarse-grained to fine-grained level.

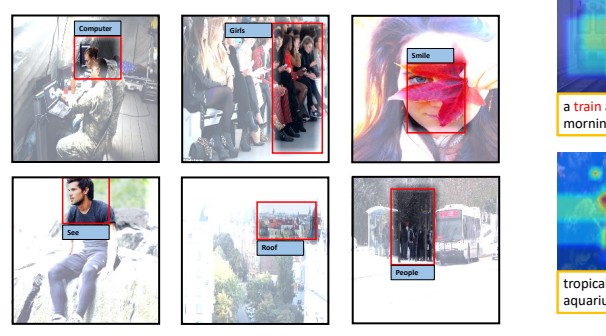

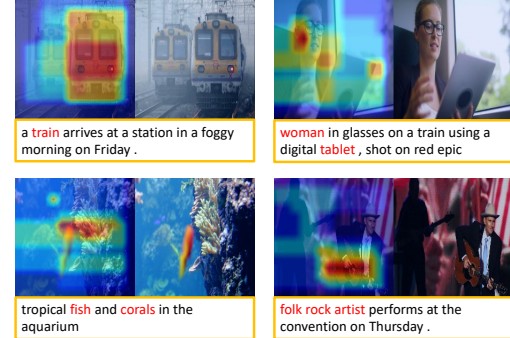

(a) Attended regions with respect to the most relevant word in the corresponding sentence.

(b) Attention heatmap of video frames and corresponding sentence. Relevant regions are highlighted.

Figure 10: Visualization of the alignment between video and language.

### A.4.8 RESULTS ON MSRVTT CAPTION.

| Method | MSRVTT | | | |
|---|---|---|---|---|
| | BLEU4 | METEOR | ROUGE-L | CIDEr |
| OA-BTG (Zhang & Peng, 2019) | 41.4 | 28.2 | - | 46.9 |
| GRU-EVE (Aafaq et al., 2019) | 38.3 | 28.4 | 60.7 | 48.1 |
| SAAT (Zheng et al., 2020) | 39.9 | 27.7 | 61.2 | **51.0** |
| STG-KD (Pan et al., 2020) | 40.5 | 28.3 | 60.9 | 47.1 |
| SupportSet (Patrick et al., 2021) | 38.9 | 28.2 | 59.8 | 48.6 |
| VideoAsMT (Korbar et al., 2020) | 41.7 | 28.5 | - | - |
| UniVL (Luo et al., 2020) | **41.8** | 28.9 | 60.8 | 50.0 |
| Ours | 40.8 | **29.0** | **61.4** | 50.2 |

Table 14: Results of MSRVTT caption.

Results are shown in Table 14. Our method outperforms other methods in terms of METEOR and ROUGE-L metrics. On BLEU4 and CIDEr, our method's results are also close to the state-of-the-art. Note that different previous works of video captioning which rely on densely sampled raw frames as input, our model only requires the light-weight object features. Results show that our method still achieve satisfying performance with less computation cost.

## A.5 CODE INSTRUCTION.

In this section, we give a step-by-step instruction to illustrate how to run the code we provided in supplementary. We first describe how to set up the python environment and prepare the checkpoints to initialize video and text encoder. Then, we introduce how to prepare the pre-training and downstream tasks, the tools to extract region features are also given. The commands to run the pre-training and downstream task, and explanation of config files are shown at the end.

---

**Algorithm 1** Code Instruction.

---

**Requirements.**
- `conda create -n demovlp python=3.8`
- `source activate demovlp`
- `pip install -r requirements`

**Download Pre-trained Model.**
- `mkdir pretrained`
- `cd pretrained`
- `wget -c https://github.com/rwightman/pytorch-image-models/releases/download/v0.1-vitjx/jx_vit_base_p16_224-80ecf9dd.pth`
- `mkdir distilbert-base-uncased`
- Download all files from huggingface distilbert-base-uncased, and put them into `pretrained/distilbert-base-uncased`.

**Pre-train Datasets.**
- WebVid: Refer to WebVid.
- CC3M: Refer to Conceptual Captions Website.

**Downstream Datasets.**
- MSRVTT Retrieval Dataset: `wget -c https://www.robots.ox.ac.uk/~maxbain/frozen-in-time/data/MSRVTT.zip`

**Extract Region Features.**
 To save time and storage consumption, we uniformly sample 8 frames for each video in WebVid and extract region features for each frame. There two ways to get region features.
- We provide a hyper-link (will release after acceptance) to download.
- Refer to the scripts in bottom-up-attention.

**Pre-train.**
- Specify `data_dir` and `object_dir` in `configs/pt/o2t-cl-local-select-loss-cc.json` to directories that contain raw videos and region features.
- `python -m torch.distributed.launch --nproc_per_node 8 --master_port 2912 train_dist_multi.py --config configs/pt/o2t-cl-local-select-loss-cc.json -sc 30 40`

**Downstream Task.**
- Specify `data_dir` and `object_dir` in `configs/ft/msrvtt_o2t-select.json-cc.json` to directories that contain raw videos and region features.
- Specify `load_checkpoint` to the pre-trained checkpoint file.
- `python -m torch.distributed.launch --nproc_per_node 2 --master_port 2912 train_dist_multi.py --config configs/ft/msrvtt_o2t-select.json -sc 2 4 8`

**Explanation of Config File.**
 We explain several important terms in the config files for easy to run the code.
- `data_dir`: the directory that contains original videos.
- `object_dir`: the directory that contains region features.
- `object_num`: the number of regions extracted from per frame.
- `num_frames`: the number of frames sampling from per video.
- `load_checkpoint`: the checkpoint file of pre-training model.

---

