# OpenReview forum: "Revitalize Region Feature for Democratizing Video-language Pre-training of Retrieval"
_ICLR.cc/2023/Conference — Submitted to ICLR 2023_

### Official Review · Reviewer_ymrc · 2022-10-23

**Confidence:** 5
**Correctness:** 4
**Technical Novelty And Significance:** 2
**Empirical Novelty And Significance:** 2
**Recommendation:** 6

**Clarity, Quality, Novelty And Reproducibility:**

This paper is clear enough to understand. The quality is good, but the novelty is marginal as mentioned in weakness. Since the code is provided by the authors, its reproducibility can be guaranteed.


**Strength And Weaknesses:**

Strength:
1. VLP requires a huge amount of computational resources, which is not affordable by a lot of researchers. Therefore, it is important to reduce the training time without sacrificing the model performance.
2. The performance is good.
3. The paper is well-organized and easy to follow.
4. Thanks for proving the code which can help readers to reproduce the results.

Weakness:
1. On page 5, what’s the motivation of this Region-Word Alignment design? Why is it better than existing local alignment methods, such as FILIP? There are lots of work that utilize optical transport (OT) to align regions and words (e.g., UNITER: UNiversal Image-TExt Representation Learning), what’s the advantage of the proposed Region-Word Alignment compared to OT?
2. To reduce the pre-training time, this paper proposes two strategies, i.e., (M1) reduce temporal visual redundancy by frame sampling and (M2) reduce spatial visual redundancy by using region features. M1 is widely used in exiting VLP methods (e.g., Align and Prompt). In terms of M2, there are some works (e.g., MAE and VideoMAE) that mask 90%-95% patches/tubes/cubes in image/frame/video to reduce the spatial visual redundancy, what’s the advantage of using region features from a pre-trained object detector? Of course, the pre-trained object detector can be used multiple times.
3. This paper uses a well-trained Faster-RCNN to extract region features then uses ViT to learn interactions between region features. Is that fair to compare with other methods which only use ViT as the video encoder?


**Summary Of The Paper:**

This paper proposes a new video-language pre-training (VLP) method which is computationally efficient. The key idea is to reduce (1) temporal visual redundancy by frame sampling and (2) spatial visual redundancy by using region features that are extracted by a pre-learned object detector. To align region features and text tokens, this paper also proposes a bidirectional region-word alignment regulation. The efficiency and performance of the proposed method is empirically proved.


**Summary Of The Review:**

The method proposed in this work demonstrates significant performance improvement in retrieval tasks. However, the technical novelty of this paper is marginal.

---

> ### Author Response · Authors · 2022-11-17
> **Responses to Reviewer ymrc (1/2)**
>
> Thank you for the constructive comments for improving our clarity of motivations and experimental verifications. The responses to your main concerns are listed below.
>
>
>
> **Q1:**
> **(1)** What’s the motivation for this Region-Word Alignment design?
>
> **(2)** Why is it better than existing local alignment methods, such as FILIP?
>
> **(3)** What’s the advantage of the proposed Region-Word Alignment compared to OT (e.g., UNITER)?
>
> **A1:** Thank you for the question.
>
> **(1)** As mentioned in the Introduction of the main paper, off-the-shelf features (e.g., region features) have been phased out in VLP tasks due to inferior performance, which was attributed to the **disconnections between the pre-extracted features and language modality** [1]. This motivates us to eliminate the disconnections by imposing proper regularizations (i.e., the bidirectional region-word alignment in our paper) with an intuition that objects from video frames are naturally associated with certain words in the corresponding sentences. State-of-the-art performances on downstream tasks demonstrate our argument that the disconnections do not pose challenges to the widespread use of region features when equipped with proper regularization.
>
> Note that, although the implementation is somewhat similar, our motivation significantly differs from previous local alignment methods. Previous works adopt local alignment for incremental gains, while the region-word alignment is crucial for our method to eliminate the disconnections, enabling a SOTA model with pre-extracted region features.
>
> > [1] Less is more: Clipbert for video-and-language learning via sparse sampling. CVPR 2021.
>
> **(2)** We discuss other local alignment methods (e.g., TACo [2], FILIP [3]) in Sec. 4.2 and the experimental results can be found in Tab. 5. Though they somewhat eliminate the disconnections and improve the baseline performance (i.e., +7.3% for TACo and +5.3% for FILIP), they are still much inferior to ours (i.e., -6.2% for TACo and -8.2% for FILIP) since **they only associate the image/region with a single word**. Intuitively, an image/region should carry combined semantics from multiple words. We properly align a region with multiple possible words via attention in Eqs. (6-7).
>
> **The above sub-optimal alignment is not critical for their end-to-end frameworks due to the incremental gains, but is fatal to our framework due to the incomplete elimination of disconnections, leaving a significant gap from SOTA results.** A similar conclusion can be drawn from Tab. 4, where we discuss the effect of region-word alignment on our framework with off-the-shelf region features and the end-to-end framework Frozen [4]. As observed, the alignment achieves +13.5% on our non-e2e framework and only +4.1% on the e2e Frozen. Interestingly, despite the significant gap initially (non-e2e 22.5% vs. e2e 31.0%), our non-e2e framework outperforms Frozen when equipped with region-word alignment (non-e2e 36.0% vs. e2e 35.1%).
>
> > [2] Taco: Token-aware cascade contrastive learning for video-text alignment. ICCV 2021.
> > [3] FILIP: Fine-grained interactive language-image pre-training. ICLR 2022.
> > [4] Frozen in time: A joint video and image encoder for end-to-end retrieval. ICCV 2021.
>
>
> **(3)** Though OT-based alignment as in UNITER theoretically overcomes the single-word issue of TACo and FILIP, **it heavily suffers from the training efficiency**. Intuitively, the OT-based alignment performs like image-text matching with a complexity of O(NxM) while ours performs like image-text contrastive with a complexity of O(N+M). Specifically, OT-based alignment computes the OT distance for each image-text pair individually, while we do it in parallel. OT-based alignment assigns only one negative for each query (1:1 positive and negative pairs) due to the limitation of high computational overhead, while we have B-1 negatives for each query given B samples in a batch. Even so, pre-training of OT-based alignment is still **2x slower** than ours per epoch. Moreover, due to the **limited negative samples**, OT-based alignment achieves inferior performance than the introduced bidirectional region-word alignment in our framework, i.e., -5.0% R@1 on MSRVTT retrieval (slightly better than FILIP and TACo). *Last but not least, the inefficiency of OT-based alignment is contrary to the democratization of vision-language pre-training we advocate.*

---

> > ### Author Response · Authors · 2022-11-17
> > **Responses to Reviewer ymrc (2/2)**
> >
> > **Q2:** What's the advantage of using region features from a pre-trained object detector to reduce spatial redundancy compared with MAE and VideoMAE?
> >
> > **A2:** Thank you for the question. We would like to first recall that the random patch masking in MAE/VideoMAE **serves for patch reconstruction** that encourages local context reasoning **rather than reducing visual redundancy**. We explain it in detail from two perspectives as follows.
> >
> > **First**, the information dropped by MAE/VideoMAE is not entirely redundant. Applying random patch masking without reconstruction leads to semantic loss and misalignment between partial visual information and full textual descriptions, i.e., -5.5% inferior than ours on MSRVTT using 90% dropout. In contrast, our region features carry salient semantics filtered by the off-the-shelf detector and effectively reduce visual redundancy almost without information loss. **Second**, applying random patch masking with reconstruction as multi-task learning (i.e., Frozen + MAE) actually results in 1.3x more training time than Frozen (7.8x than ours) per epoch due to the requirement of two-pass forward towards two training objectives. This is contrary to the democratization of vision-language pre-training we advocate.
> >
> > > [1] Frozen in time: A joint video and image encoder for end-to-end retrieval. ICCV 2021.
> >
> >
> >
> > **Q3:** Is that fair to compare with other methods which only use ViT as the video encoder?
> >
> > **A3:** Thank you for the question. Using pre-trained models to extract features is actually a common practice in VLP. Taking methods we compared in Tab. 7 as examples, ActBERT [1] introduces a ResNet3D pre-trained on Kinetics 400 to extract video features. MMT [2], MDMMT [3] and CE [4] use multiple experts pre-trained on different datasets (e.g., Kinetics 400, Places365, ImageNet) to extract offline features for pre-training. Please also note we show lower FLOPs for detector+ViT due to the much lower computational complexity in sparse self-attention, compared to end-to-end methods that only use ViT (see Tab. 3). Thus, the comparison in our paper is fair.
> >
> > > [1] Actbert: Learning global-local video-text representation. CVPR 2020.
> > > [2] Multi-modal transformer for video retrieval. ECCV 2020.
> > > [3] Mdmmt: Multidomain multimodal transformer for video retrieval. CVPR 2021.
> > > [4] Use what you have: Video retrieval using representations from collaborative experts. BMVC 2019.
> >
> >
> >
> >
> > **Q4:** The technical novelty of this paper is marginal.
> >
> > **A4:** Though the individual techniques (region features, local alignment) seem not new, we have **a clear emerging motivation** of democratizing VLP research and **offer a practical solution** for this, saving 80% data and 85% pre-training time compared to SOTA. Also note that our solution is not simply composed of existing techniques, but is **specifically designed for our task**.
> >
> > (1) Regarding the motivation for democratizing VLP: Existing VLP methods are quite data-hungry due to a large number of model parameters and uncurated raw inputs. Their pre-training stage turns out to be expensive with massive pre-training data and long pre-training time, making it unaffordable for most researchers to study VLP. Our goal is to propose an extremely efficient and powerful VLP framework with SOTA results, democratizing VLP to more researchers to contribute to the community. (2) Regarding the motivation for using region features rather than patch masking to reduce visual redundancy, please see A2 for details. (3) Regarding the motivation and advantages for our region-word alignment, please see A1 for details.

---

### Official Review · Reviewer_nWq8 · 2022-10-24

**Confidence:** 5
**Clarity, Quality, Novelty And Reproducibility:** The idea is clear and easy to underst…
**Correctness:** 3
**Technical Novelty And Significance:** 2
**Empirical Novelty And Significance:** 2
**Recommendation:** 3

**Strength And Weaknesses:**

Strengths:
1. The paper is well-written and easy to understand. The presentation is clear.
2. The authors proposed an efficient pre-training approach for video-language tasks. Some discussions regarding video temporal and spatial redundancy are interesting.
3. The authors provide thorough experiments and show that their pre-training is faster than prior works.
4. The pre-training objective, including video-sentence alignment and region-word alignment, is technically valid.

Weaknesses:
1. The proposed method heavily relies on pre-trained object detector. It seems very challenging and not affordable to extract/store all the region features, especially for large-scale video datasets. Such solution seems not appealing as the object detectors are keep improving. The region features need to be re-extracted if newer object detector is available. The feature extraction is very time consuming. Feature storage is very expensive. All video files need to be stored in the system for feature extraction if new object detector is available.
2. The proposed method use mixture of video-text and image-text data for pre-training. But it is unclear what is the pre-training objective for the image-text inputs. I wonder if the proposed pre-training mainly learns from image-text inputs or video-text inputs.
3. The authors only conduct experiments on retrieval tasks. It is unclear if the pre-training is still useful when it comes to video QA and video captioning.

**Summary Of The Paper:**

This paper presents an efficient video-language pre-training method. The key idea is to employ region-text pairs during the pre-training. Different from recent works that use end-to-end transformers, the authors revisit the object region features extracted from pre-trained object detector. The idea is somehow similar to OSCAR, but extending such use of region-text pairs for video-language problem. Experimental results show that their method achieves better performance on downstream image-text retrieval.

**Summary Of The Review:**

Overall, the proposed method achieves nice improvements compared to previous SOTA on image-text retrieval. This shows that prior end-to-end methods have a lot more spaces to improve. However, the proposed method seems very expensive in terms of feature extraction and feature storage. It may have difficulty to scale. In addition, as a pre-training paper, it would be good to verify their method on multiple downstream tasks. Now, the paper only considers retrieval, which is a bit narrow.

---

> ### Author Response · Authors · 2022-11-17
> **Responses to Reviewer nWq8 (1/3)**
>
> Thank you for the constructive comments for improving our clarity of statements and experimental verifications. The responses to your main concerns are listed below.
>
>
>
> **Q1:** The idea is somehow similar to OSCAR, but extending such use of region-text pairs for video-language problem.
>
> **A1:** We sincerely appreciate the reviewer's comments, but regret that the reviewer confused our work with previous ones that employ detectors (e.g., OSCAR) while neglecting our main motivation, that is, **making VLP research affordable for more researchers (dubbed “democratization” in our paper)**. Such a motivation emerges from the increasing requirement for data and training resources for current VLP research, making it quite expensive to pursue research in this direction. The motivation is new and we offer a practical solution for this, saving 80% data and 85% pre-training time compared to SOTA.
>
>
> **Q2:** The proposed method heavily relies on pre-trained object detector.
>
> **A2:** We do not think such “reliance” constitutes a weakness since (i) we did not rely on a powerful detector to achieve SOTA results (also not our motivation), and (ii) we did not increase pre-training and inference time due to the use of the detector.
>
> Rather than achieving incremental gains with a strong detector, we aim to verify the claim that **properly using region features as input significantly reduces the training resources without performance loss, towards democratizing VLP research for most researchers**. As for the detector we used in our experiments, we directly follow the widely-used one in ActBERT [1], UNITER [2], and OSCAR [3]. Of course, it can be replaced by any available ones in practice.
>
> By properly leveraging region features instead of raw pixels as input, we save 80% of data and 85% pre-training time compared to the most efficient SOTA method while achieving competitive results. As for inference, we show lower FLOPs for detector+ViT due to the much lower computational complexity in sparse self-attention, compared to end-to-end methods (see Tab. 3). Namely, we do not need extra time for building index features.
>
> Moreover, it is quite common to leverage off-the-shelf models to extract features in VLP research as well as real-world problems. Taking methods we compared in Tab. 7 as examples, ActBERT [1] introduces a ResNet3D pre-trained on Kinetics 400 to extract video features. MMT [4], MDMMT [5] and CE [6] use multiple experts pre-trained on different datasets (e.g., Kinetics 400, Places365, ImageNet) to extract offline features for pre-training.
>
> Please see below Q3 for the reviewer’s main concerns on the practicality of object detectors.
>
> > [1] Actbert: Learning global-local video-text representation. CVPR 2020.
> > [2] UNITER: UNiversal Image-TExt Representation Learning. ECCV 2020.
> > [3] Oscar: Object-Semantics Aligned Pre-training for Vision-Language Tasks. ECCV 2020.
> > [4] Multi-modal transformer for video retrieval. ECCV 2020.
> > [5] Mdmmt: Multidomain multimodal transformer for video retrieval. CVPR 2021.
> > [6] Use what you have: Video retrieval using representations from collaborative experts. BMVC 2019.

---

> > ### Author Response · Authors · 2022-11-17
> > **Responses to Reviewer nWq8 (2/3)**
> >
> > **Q3:**
> > (1) The region features need to be re-extracted if a newer object detector is available.
> >
> > (2) The feature extraction is very time consuming.
> >
> > (3) Feature storage is very expensive.
> >
> > (4) All video files need to be stored in the system for feature extraction if a new object detector is available.
> >
> >
> > **A3:**
> > (1) We agree that better object detectors bring better performance, however, almost all VLP methods would benefit from the improvement of video and text encoders. It is quite common to retrain a model when a stronger architecture or better model weights are available. We hold the opinion that this should not be a specific problem with our method. If the reviewer is concerned about the extra time and storage required by region feature extraction, please see below for details.
> >
> > (2) We save extraction time via sparse sampling. As shown in Tab. 1, pre-training with only 20% videos can achieve SOTA results. It takes about 100ms for each video to extract region features using a sole V100, as shown in Tab. 2. It takes us ~1.39 days to sequentially extract region features for 20% of the video data (1.2M) using a single V100, which can be further speed-up using multiple GPUs for extracting in parallel. Moreover, the feature extraction process is disposable for algorithm verification. We will provide extracted features for the community to directly download for algorithm development. As for deployment, detector+ViT are required to build the index end-to-end. As shown in Tab. 3, due to the lower FLOPs of detector+ViT, we do not require extra time compared to end-to-end methods.
> >
> > (3) Sparse sampling also reduces space costs. Given that WebVid-2.5M requires 4.7T for saving raw videos, we require 0.275T storage for 1 frame per video and 1.1T for 4 frames per video, when using 100% video data. Moreover, if we extremely use 20% of videos with 1 frame, we only require 0.055T (56G) for saving features. Note that, as shown in Tab. 1, pre-training with either 1 frame or 4 frames consistently achieves SOTA performance. Given the storage cost on AWS ($0.023 per GB), it takes us only about 25 dollars a month to save the 4-frame region features for full WebVid-2.5M.
> >
> > (4) Note that saving raw video data for feature upgradation is required by many previous works, including those detector-based and also the CLIP-based, which is not a specific problem of our work.
> >
> >
> > **Q4:** (1) It is unclear what is the pre-training objective for the image-text inputs. (2) If the proposed pre-training mainly learns from image-text inputs or video-text inputs.
> >
> > **A4:** Combining image-text and video-text pairs for video-text pre-training is common in recent works, e.g., Frozen [1].
> > (1) We treat images as one-frame videos to perform the same pre-training objective as video-text inputs, following the practice in Frozen.
> > (2) We acknowledge that image data is useful since image-text datasets carry more high-quality image-text pairs with clear literal descriptions and diverse visual signals. As shown in Tab. 13 of Appendix, compared with the default setting (i.e., WebVid-2.5M+CC3M), R@1 on MSRVTT retrieval drops -8.4% when discarding CC3M. It is hard to say which is more important due to the inconsistent data magnitude for image and video datasets. But we can confirm that temporal modeling is crucial for finetuning given the significant performance drops when using only 1 frame as in Fig. 7.
> >
> > > [1] Frozen in time: A joint video and image encoder for end-to-end retrieval. ICCV 2021.

---

> > > ### Author Response · Authors · 2022-11-17
> > > **Responses to Reviewer nWq8 (3/3)**
> > >
> > > **Q5:** It is unclear if the pre-training is still useful when it comes to video QA and video captioning.
> > >
> > > **A5:** We provided the results on multiple video QA benchmarks in A.4.2 and A.4.3 of the Appendix in the initial version. The results further verify the effectiveness of our model. We will complement captioning experiments in the final version.
> > >
> > > Note that our approach is a dual-encoder structure specially designed for video-language pre-training for retrieval tasks (also pointed out in our title), given that QA and captioning rely on a single-stream fusion module. The verification on QA/captioning can be considered a bonus and it is also common to design methods for only retrieval tasks in video-language pre-training, e.g., Frozen [1], OA-Trans [2].
> > >
> > > > [1] Frozen in time: A joint video and image encoder for end-to-end retrieval. ICCV 2021.
> > > > [2] Object-aware Video-language Pre-training for Retrieval. CVPR 2022.
> > >
> > >
> > >
> > >
> > > **Q6:** It may be difficult to scale.
> > >
> > > **A6:** In Appendix.4.6 (Tab. 13) of our initial version, we have shown the results of our method with different data scales (copy below for your convenience). As the data scale up, our method obtains better results. This verifies that our method can not only achieve SOTA results with much fewer data, but is also scalable with more data.
> > >
> > >
> > > |       Data       | Data Scale |  R@1 |  R@5 | R@10 |
> > > |:----------------:|:---------------:|:--------:|:--------:|:--------:|
> > > |      WebVid   |    2.5M        | 27.6    | 50.4   | 61.7   |
> > > |    WebVid+CC3M   |    5.8M    | 36.0 | 61.0 | 71.8 |
> > > |     CC3M+CC7M    |    10.3M   | 38.7 | 63.8 | 73.9 |
> > > | WebVid+CC3M+CC7M |    12.8M   | 39.6 | 64.8 | 73.8 |

---

> > > > ### Author Response · Authors · 2022-11-18
> > > > **Added captioning results, looking forward to your reply**
> > > >
> > > > Dear Reviewer nWq8,
> > > >
> > > > We are writing to kindly inform you of the newly added results on the downstream captioning task (see A.4.8 of the Appendix), which consistently indicates the effectiveness of our method. Also, see A.4.2 and A.4.3 for QA tasks.
> > > >
> > > > Regarding your main concern on the practicality of object detectors, we clarified that (1) the feature extraction is actually not expensive and disposable for algorithm verification, (2) the feature storage is also affordable for most ones (25 dollars/month for WebVid-2.5M), (3) the feature extraction does not increase inference time for deployment compared to end-to-end methods due to the fewer FLOPs. Please see A2-3 for details.
> > > >
> > > > We are wondering if our responses above and additional experiments have well addressed your concerns. Looking forward to your reply and the discussions.
> > > >
> > > > Best Regards,
> > > >
> > > > ICLR 2023 Conference Paper1802 Authors

---

> > > > > ### Comment · Reviewer_nWq8 · 2022-11-28
> > > > > **Response**
> > > > >
> > > > > Thank you for the additional explanations and experiments. However, it doesn't convince me well. The method is mainly relied on sparse sampling to reduce the costs. As discussed in many recent papers; sparse sampling can somehow hack existing video-language downstream datasets. This is because those videos are just the short video clips. When it comes to unconstrained videos in real-world applications, it is hard to convince me that sparse sampling is able to learn the detailed motions, actions, and high-level semantic understanding. If we sample more video frames, then the computational cost of the proposed method may not be affordable to most researchers. It is hard to convince me that the proposed method is the future we should pursue.

---

> > > > > > ### Author Response · Authors · 2022-11-28
> > > > > > **Response to Reviewer nWq8**
> > > > > >
> > > > > > We sincerely appreciate the reviewer’s response. Regarding the reviewer’s new concern about the feasibility of sparse sampling, we would like to respond from the following perspectives.
> > > > > >
> > > > > > +  Sparse sampling has been proven to be effective in state-of-the-art video-text pre-training methods, including ClipBERT [1], Frozen [2]. Similar conclusion has also been drawn from video self-supervised learning methods (e.g., VideoMAE [3]) that consecutive video frames are highly redundant and the semantics vary slowly in the temporal dimension. The practice of sparse sampling is grounded and not first introduced in our work.
> > > > > > + Following the common practice (as in ClipBERT [1]), we only adopt sparse sampling during pre-training while densely sampling more frames for downstream tasks. The detailed information can be learned during downstream fine-tuning. And well noted that it is very expensive to perform pre-training with densely sampled video frames even using an end-to-end framework, especially when the data are scale-up.
> > > > > > + We agree with the reviewer that the current VLP methods might be biased by short video clips and infeasible for unconstrained video understanding. However, it should not be a specific problem with our method but a common problem for most VLP models (e.g., ClipBERT [1], Frozen [2]) pre-trained with short video clips (e.g., WebVid-2.5M). Moreover, it should not be totally denied that there exist applications with short videos in the real world.
> > > > > >
> > > > > > As last, we would like to recall that our democratization of VLP pre-training on the widely-used datasets of short video clips is a practical contribution to the community. It is undeniable that it has application limitations (we will point it out in the conclusion section), such as unconstrained video understanding, however, it is actually another research topic with different benchmarks.
> > > > > >
> > > > > > > [1] Less is More: ClipBERT for Video-and-Language Learning via Sparse Sampling. CVPR 2022 (Best Student Paper Honorable Mention).
> > > > > > > [2] Frozen in time: A joint video and image encoder for end-to-end retrieval. ICCV 2021.
> > > > > > > [3] VideoMAE: Masked Autoencoders are Data-Efficient Learners for Self-Supervised Video Pre-Training. NeurIPS 2022.

---

### Official Review · Reviewer_yXUD · 2022-10-25

**Confidence:** 3
**Correctness:** 3
**Technical Novelty And Significance:** 2
**Empirical Novelty And Significance:** Not applicable
**Recommendation:** 5

**Clarity, Quality, Novelty And Reproducibility:**

See "Strength And Weaknesses" for clarity, quality and reproducibility.

In terms of novelty, I think most of the components have been used before, though for the practical types of papers I guess it does not matter that much.



**Strength And Weaknesses:**

Strengths:
1) The approach allows getting a higher performance compared to the baselines. Pretrained detector allows to directly inject information from the labeled detection datasets, plus word alignment helps improve the model convergence.
2) The general idea and solution are presented quite clearly in the paper.

Weaknesses:
1) No instructions on how to run the code and reproduce the results. This is critical as the paper claims "democratization" of the video retrieval.
2) The paper writing might be improved. There are several references to the equation (10) in the paper, while it is actually in the supplementary, which does not make much sense. The tables 4,5,6 are not clear, particularly what is "base". The section 3.3. is hard to follow, the equations lack a word explanation of why they intend to do, there is also no corresponding illustration for those.
3) The detector network is old and not trained end-to-end. It is mentioned in the text there are no bounding-boxes provided for training, but there is also an option to just backpropogate the final training loss back to the backbone on some architectures.


**Summary Of The Paper:**

The paper proposes a method for video-language retrieval. The method is based on pretraining a general still image detector and feeding the pooled output of the detector plus position+time features into a transformer to produce a video embedding. A global loss on cls video embedding, as well as local individual token losses (reminiscent of ColBert) are used to train the model. The approach is benchmarked on several datasets and exhibited better performance compared to the baselines and used less pretraining time (though that probably does not include the detector pretraining).

**Summary Of The Review:**

It does not seem like the paper offers a big breakthrough in terms of ideas, but the focus of the paper is pretty clear, so novelty does not seem be an issue.
I think for democratization of the video retrieval the major ingredient is the reproducible code that is easy to run. The submission does have a code associated with it, but it lacks instructions on how to run it, which creates artificial difficulties for a review.
The paper also has issues with presentation discussed in the weaknesses, so overall I vote for slightly below the acceptance threshold.

---

> ### Author Response · Authors · 2022-11-17
> **Responses to Reviewer yXUD**
>
> Thank you for the constructive comments for improving our writing and experimental verifications. The responses to your main concerns are listed below.
>
>
> **Q1:** No instructions on how to run the code and reproduce the results. It is critical for "democratization" of the video retrieval.
>
> **A1:** Thank you for the advice. We have complemented more detailed descriptions in Appendix A.5 to illustrate how to run the code. Meanwhile, we add a README file in the code. We will also constantly iterate the codebase for ease of use.
>
> However, we would like to clarify that **democratizing VLP research is not limited to open-source code, but to make VLP research affordable for more researchers**, i.e., achieve SOTA results in a cheap way. Such a motivation emerges from the increasing requirement for data and training resources for current VLP research, making it quite expensive to pursue research in this direction. We offer a practical solution for this, saving 80% data and 85% pre-training time compared to SOTA methods.
>
>
> **Q2:** The writing issues.
>
> **A2:** Thank you for the helpful suggestions. We have revised our manuscript accordingly in red. (1) We revised the reference from equation (10) to equation (7) in our main paper, which are symmetric formulas with similar meanings. (2) We explained more about the meaning of “base” in the caption of Tabs. 4/5. (3) We rewrite Sec. 3.3 with more intuitions for easier understanding and add an illustration of the region-word alignment in Fig. 8 of the Appendix. We will also carefully proofread the paper for the final version.
>
>
> **Q3:** The detector network is old and not trained end-to-end. It is mentioned in the text there are no bounding-boxes provided for training, but there is also an option to just backpropagate the final training loss back to the backbone on some architectures.
>
> **A3:**
> **Regarding “old”:** As for the detector we used in our experiments, we directly follow the widely-used one in ActBERT [1], UNITER [2], and OSCAR [3]. Of course, it can be replaced by any available ones in practice. However, we did not intend to achieve state-of-the-art performance with the reliance on an up-to-date detector. We aim to verify our argument that properly using region features as input significantly reduces the training resources without performance loss, towards democratizing VLP research for most researchers.
>
> **Regarding “end-to-end”:** Actually, it is infeasible to “democratize” VLP by backpropagating back to the detector backbone since it requires much more GPU memory to include a detector network for joint training. Instead, we extract region features once and efficiently develop algorithms directly with the input of region features.
>
> According to the reviewer’s suggestion, we attempted to backpropagate the loss to the detector without any ground-truth of bounding-boxes. But unfortunately, the training collapsed with a loss of nan from the second epoch, probably due to the trivial solutions of region proposals. Perhaps careful tuning of hyperparameters can stabilize training, but for our work on democratization, this is not necessary due to the additional GPU memory required.
>
>
> > [1] Actbert: Learning global-local video-text representation. CVPR 2020.
> > [2] UNITER: UNiversal Image-TExt Representation Learning. ECCV 2020.
> > [3] Oscar: Object-Semantics Aligned Pre-training for Vision-Language Tasks. ECCV 2020.

---

> > ### Author Response · Authors · 2022-11-18
> > **Looking forward to your reply and the discussions**
> >
> > Dear Reviewer yXUD,
> >
> > Thank you again for your acknowledgment of the practicality of our work. Regarding your main concerns in the initial comments, we (1) provided instructions for our released code to ensure reproducibility, (2) proofread our paper and added more intuitions and illustrations to improve readability, (3) attempted end-to-end training and explained its infeasibility on democratizing VLP.
> >
> > In terms of your concerns on the limited novelty, we would like to claim that though the individual components (region features, local alignment) seem not new, we have a clear emerging motivation of democratizing VLP research (making it affordable for most ones) and offer a practical solution for this, saving 80% data and 85% pre-training time compared to the most efficient SOTA. Also note that our solution is not simply composed of existing techniques, but is specifically designed for our task.
> >
> > We are wondering if our responses have well addressed your concerns. Looking forward to your reply and the discussions!
> >
> > Best Regards,
> >
> > ICLR 2023 Conference Paper1802 Authors

---

### Decision · Program_Chairs · 2023-01-20

**Decision:**

Reject

**Justification For Why Not Higher Score:**

While the paper does propose a computationally efficient model, it does not appear to be able to provide competitive performance compared to the most recent VLP models (this field is very dynamic). Given this and the relative lack of novelty, I do not see a lot of insights this paper can bring to the already densely populated, in terms of publishing, field.

**Justification For Why Not Lower Score:**

N/A

**Metareview: Summary, Strengths And Weaknesses:**

The paper proposes an approach for video-language retrieval (and in supplementals Video QA). The approach uses a detector features from which are leveraged in a transformer with a novel bidirectional region-word alignment regularization. Paper was reviewed by three reviewers and received the following ratings:

* 1 x reject, not good enough
* 1 x marginally below the acceptance threshold
* 1 x marginally above the acceptance threshold

No reviewer strongly argued for the paper. Concerns of reviewers were multi-fold: (1) limited novelty [yXUD, nWq8, ymrc], (2) issues with presentation and exposition [yXUD] and (3) limited evaluation on only one/few downstream tasks [nWq8]. It was also unclear if the pre-traded object detector time was counted in benchmarking.

Authors have provided a rebuttal that tried to address some of these concerns. Reviewers have not commented on these argument and have not upgraded the scores. AC has carefully looked at the reviews, rebuttal and paper itself. Overall, AC agrees with limitations stated by the reviewers and does not believe rebuttal addresses aforementioned limitations well. Specifically:

* The novelty of proposed approach is indeed very limited. Other methods have proposed similar region-word alignment objectives. Even though proposed formulation is different, it does not fundamentally add any deep insights.

* The performance while is indeed faster than prior works, cannot match the computational performance of more recent architectures (e.g., MERLOT, VIOLET and a number of others). Also with respect to those architectures, performance is shown on fewer tasks.

* The issues with presentation and exposition, while improved in the revision, are not completely relinquished.

Based on these and other factors mentioned by the reviewers, the paper does not appear to be ready for publication at this time.